# The Contribution of the Intestinal Microbiota to the Celiac Disease Pathogenesis along with the Effectiveness of Probiotic Therapy

**DOI:** 10.3390/microorganisms11122848

**Published:** 2023-11-23

**Authors:** Oxana Zolnikova, Natiya Dzhakhaya, Elena Bueverova, Alla Sedova, Anastasia Kurbatova, Kira Kryuchkova, Tatyana Butkova, Alexander Izotov, Ludmila Kulikova, Kseniya Yurku, Pavel Chekulaev, Victoria Zaborova

**Affiliations:** 1Institute of Clinical Medicine, I.M. Sechenov First Moscow State Medical University (Sechenov University), 119991 Moscow, Russia; zolnikova_o_yu@staff.sechenov.ru (O.Z.); dzhakhaya_n_l@staff.sechenov.ru (N.D.); bueverova_e_l@staff.sechenov.ru (E.B.); sedova_a_v@staff.sechenov.ru (A.S.); kurbatova_a_a@staff.sechenov.ru (A.K.); chekulaev_p_a@student.sechenov.ru (P.C.); 2Institute of Public Health, I.M. Sechenov First Moscow State Medical University (Sechenov University), 119991 Moscow, Russia; kryuchkova_k_yu@staff.sechenov.ru; 3Institute of Biomedical Chemistry, Biobanking Group, 109028 Moscow, Russia; t.butkova@gmail.com (T.B.); izotov.alexander.ibmc@gmail.com (A.I.); likulikova@mail.ru (L.K.); 4Institute of Mathematical Problems of Biology RAS—The Branch of Keldysh Institute of Applied Mathematics of Russian Academy of Sciences, 142290 Pushchino, Russia; 5State Research Center—Burnasyan Federal Medical Biophysical Center of Federal Medical Biological Agency, 123098 Moscow, Russia; ks_yurku@mail.ru

**Keywords:** celiac disease, gluten, gluten enteropathy, gut microbiota, immune response, intestinal permeability, probiotics

## Abstract

The development of many human disorders, including celiac disease (CD), is thought to be influenced by the microbiota of the gastrointestinal tract and its metabolites, according to current research. This study’s goal was to provide a concise summary of the information on the contribution of the intestinal microbiota to the CD pathogenesis, which was actively addressed while examining the reported pathogenesis of celiac disease (CD). We assumed that a change in gluten tolerance is formed under the influence of a number of different factors, including genetic predisposition and environmental factors. In related investigations, researchers have paid increasing attention to the study of disturbances in the composition of the intestinal microbiota and its functional activity in CD. A key finding of our review is that the intestinal microbiota has gluten-degrading properties, which, in turn, may have a protective effect on the development of CD. The intestinal microbiota contributes to maintaining the integrity of the intestinal barrier, preventing the formation of a “leaky” intestine. On the contrary, a change in the composition of the microbiota can act as a significant link in the pathogenesis of gluten intolerance and exacerbate the course of the disease. The possibility of modulating the composition of the microbiota by prescribing probiotic preparations is being considered. The effectiveness of the use of probiotics containing *Lactobacillus* and *Bifidobacterium* bacteria in experimental and clinical studies as a preventive and therapeutic agent has been documented.

## 1. Introduction

### 1.1. General Aspects of Gut Microbiota

Most bacteria inhabiting the human body live in the intestine, accounting for about 70% of all representatives of the human bacterial flora [1,2,3]. Apparently, the formation of the microbiota occurs during intrauterine development. This hypothesis is increasingly confirmed, as bacterial communities have been identified in the placenta and meconium, previously considered sterile [4]. The main formation of the intestinal biotope occurs in the early postnatal period and is influenced by a large number of factors (method of delivery, nutritional characteristics, medication intake, physical activity, region of residence, etc.) [5]. Many authors suggest that the method of delivery plays a key role in this process [6]. It was found that infants born by caesarean section, as a rule, have more species of *Staphylococcus*, *Bacillales, Propionobacterineae, Corynebacterineae, Firmicutes* and *Acinetobacter* in the intestinal microflora, with fewer *Actinobacteria* and *Bacteroidetes*, while in the case of natural childbirth, increased colonization of the intestinal biotope *Clostridium* is revealed [7]. *Clostridia*, subsequently, actively metabolize dietary fibers into short-chain fatty acids, which have systemic anti-inflammatory effects, as will be described [8]. The development of allergic diseases, including bronchial asthma, is more common among children born by caesarean section, amounting to about 9.5%, whereas among children born naturally, the incidence is about 7.9% [9]. A number of clinical studies have shown that the intake of *Lactobacillus rhamnosus GG* and *Lactobacillus fermentum* by pregnant and lactating women in the prenatal and early postnatal periods can be effective in the treatment and prevention of atopic diseases in children [10,11].

In the human intestinal microflora, 80–90% of bacterial species belong to the types *Bacteroidetes*, *Firmicutes*, *Proteobacteria*, *Actinobacteria*, *Fusobacteria* and *Verrucomicrobia*. The types *Cyanobacteria Lentisphaerae*, *Spirochaetes* and *Synergistetes* are present in the intestinal biotope in smaller quantities [12]. The bacterial composition of different parts of the intestine is specific [13,14,15]. The stomach, duodenum and proximal small intestine are mainly colonized by aerobic bacteria, including Streptococcus, Lactobacillus and Enterobacteriaceae, while the distal part of the small intestine and colon are dominated by anaerobes such as *Bacteroides*, *Bifidobacterium*, *Prevotellaceae*, *Rikenellaceae*, *Lachnospiraceae*, *Ruminococcaceae* and *Clostridium* [16]. The resident microflora participates in maintaining the physico-chemical parameters of enterocyte homeostasis, including pH and the redox potential of the intraluminal medium.

The possibility of dividing the intestinal biotope into enterotypes is discussed in the literature [17,18]. However, this hypothesis has fewer and fewer supporters and is often criticized. The hopes pinned on the use of the enterotype for the diagnosis and prognosis of diseases have not been confirmed at present [19,20,21]. To a greater extent, the literature discusses the existence of a philometabolic microbiota core, which includes about 80 main bacteria capable of maintaining functional homeostasis of microflora [22]. However, this hypothesis can also be criticized due to the difficulty of isolating the “microbiota core” given the large fluctuations depending on genetic characteristics, age, gender, lifestyle and environmental factors [23]. The main provisions that are now defined in relation to the characteristics of the intestinal biotope relate to its species diversity, relative stability and so-called functional redundancy, especially when similar metabolic functions are performed by phylogenetically different bacteria [1,24]. All of the above, according to researchers, allows us to maintain the functional stability of biotopes.

Currently, bacteria inhabiting a particular biotope are considered as a single integral community [1,25,26]. The interaction among microorganisms within a community dictates their behavior and the direction of the biochemical processes triggered by them. Relatively recently, it has been proposed that we should term inter-bacterial interactions “Quorum Sensing” [1]. The quorum sensing system involves the synthesis of biologically active substances and metabolites by bacteria, which allows them to “communicate” at different levels: intraspecific, interspecific and for interaction with the host organism. To date, about 25,000 microbial low-molecular compounds are known to serve as effectors, cofactors and signaling molecules that regulate the rate and severity of various physiological functions [26].

### 1.2. General Aspects of Celiac Disease

Celiac disease’s etiology involves a reaction of the body to a certain component of food, namely gluten, through T cells. There are also numerous other variables contributing to the onset of this disease [27]. As a result, immuno-conditioned enteropathy develops. A feature of the intestinal lesion is the reversible atrophy of its villi and hyperplasia of crypts with the exclusion of a trigger agent. The prevalence of celiac disease according to various data ranges from 1:100 to 1:300 in the world; the ratio of women to men is 2:1 and it is higher in children than in adults (0.9% versus 0.5%) [28,29,30,31,32]. Celiac disease is more common in girls and can occur after eating gluten-containing foods at any age, including infancy [33]. Due to the nonspecificity of symptoms and the unclear clinical picture, the diagnosis is often missed even in developed countries [28]. Of course, in underdeveloped countries, diagnosis is even worse, which is a consequence of limited access to diagnostic tests and lack of experience [34].

Taking into account the clinical picture and the results of laboratory studies, two forms of the disease are distinguished: symptomatic (manifest) and asymptomatic (latent) [27,35,36]. The main manifestations of celiac disease are characterized by the presence of gastrointestinal symptoms (persistent diarrhea, flatulence, abdominal pain, nausea and others) and/or extraintestinal manifestations (osteoporosis, anemia, infertility, neurological symptoms and others). In the case of the latent form, the diagnosis is established during screening examinations. The first symptoms of the disease usually appear in childhood, 1.5–2 months after the start of consumption of gluten-containing products [37].

Gluten, as the main initiating factor of the disease, is a protein contained in cereal grains, which, along with starch, is present in their composition and consists of glutelins and prolamins. It has been confirmed that the amino acid structure of glutelin and prolamin in wheat, barley and rye is the most immunogenic for patients suffering from celiac disease [38,39]. Apparently, the change in gluten sensitivity is formed as a result of the influence of many different factors. On one hand, there is information that celiac disease develops in the presence of a genetic predisposition associated with certain human leukocyte antigen (HLA) genes of the second type, known as DQ2 and DQ8 [40,41]. However, this genotype is common and occurs in about 35% of the population, though only 3% of these people have this disease. Autoantigen tissue transglutaminase (TG) is also involved in the pathogenesis of celiac disease [42]. This indicates a possible and compelling role of other factors in the development of celiac disease, including disorders in the composition of the intestinal microbiota and increased permeability of the intestinal wall [43,44,45,46]. Given that celiac disease is characterized by inflammation that occurs in the small intestine, it can be assumed that the local microenvironment, which is significantly influenced by the microbiota, plays a decisive role in the pathogenesis of the disease and the violation of tolerance to dietary gluten. Intestinal microflora can influence the development of celiac disease through various mechanisms [47]. The microbiota, due to the peptidases secreted by it, is capable of both forming immunogenic peptides and eliminating immunogenic peptides that are not cleaved by intestinal enzymes. Some bacteria are able to express epitopes with a structure similar to gliadin, thereby triggering an immune response in the host [24]. At the same time, the microbiota can influence the formation of antigen by modulating the digestive process, generating either immunogenic or tolerant gluten peptides. In addition, the microbiome can directly affect intestinal permeability. Intestinal microbes are also involved in the regulation of immune responses, producing peptides, metabolites and cytokines that have both proinflammatory and anti-inflammatory properties [48].

## 2. The Role of the Gut Microbiota in the Pathogenesis of Celiac Disease

### 2.1. Genetically Susceptible

Several studies have revealed that infants with HLA-DQ2 and HLA-DQ8 genotypes, as well as first-degree relatives of patients with celiac disease, have a change in the composition of bacteria in the intestinal microbiota [49]. This change is expressed in an increase in the representatives of the *Firmicutes* and *Proteobacteria* groups, as well as a decrease in the number of *Actinobacteria*. The data obtained suggest that the HLA genotype is associated with certain changes in the composition of the intestinal microflora, which are characteristic of patients with celiac disease and their close relatives [50,51,52,53,54,55].

Several studies indicate a higher incidence of celiac disease among children born by caesarean section, as well as among those who are artificially fed and have also received antibacterial drugs in the first year of life [51,56,57,58,59,60,61,62]. These facts are known to affect the composition of the intestinal microbiota, which confirms the hypothesis of its significant role in the pathogenesis of this disease.

To study possible factors contributing to the risk of developing celiac disease, a research group led by Canova C. conducted a population cohort study involving 203,000 children born in northeastern Italy. In this study, 1227 children (0.6%) were diagnosed with celiac disease, of which the diagnosis was confirmed histologically in 866 cases (71%). Data analysis showed that girls had a higher risk of developing celiac disease (incidence ratio (IRR) = 1.69, 95% confidence interval (CI): 1.51, 1.90). The next significant factor, according to the authors, was infections that required hospitalization in the first year of life (IRR = 1.39, 95% CI: 1.01, 1.91), and in cases of intestinal infections, the risk of developing gluten enteropathy doubled (IRR = 2.04, 95% CI: 1.30, 3.22). It was also found that the use of antibiotics was associated with the onset of the disease (IRR = 1.24, 95% CI: 1.07, 1.43), and a dose-dependent effect was observed over several courses. Various antibiotics were compared, including penicillins, macrolides and cephalosporins. The authors noted that prescribing the latest antibiotics is associated with an increased risk of developing celiac disease (IRR = 1.42, 95% CI: 1.18, 1.73) [61].

Interesting results were obtained in the study of breast milk samples from mothers suffering from celiac disease. These mothers were found to have lower levels of interleukin 12p70, transforming growth factor-β1 and secretory immunoglobulin A (sIgA, secretory immunoglobulin A). A decrease in the amount of *Bifidobacterium* and *Bacteroides fragilis* in breast milk was also observed. This study confirms the hypothesis that a decrease in the level of immunoprotective molecules and certain types of *Bifidobacterium* can reduce the protective properties that breastfeeding usually provides, and, consequently, increase the risk of a child developing celiac disease [62,63].

### 2.2. The Composition of the Microbiota

Comparison of the composition and functional activity of bacterial communities in patients with celiac disease and healthy volunteers makes it possible to better understand the contribution of microorganisms to the development of the disease. In the case of celiac disease, changes in the composition of the microbiota are observed: the proportion of *Bacteroidetes* and *Proteobacteria* increases, and the content of *Lactobacillus*, *Bifidobacterium* and *Faecalibacterium prausnitzii* decreases [50,64,65,66,67,68,69]. Differences in the composition of *Lactobacillus* were found in children with celiac disease (predominance of *L. curvatus*) as compared to healthy children (*L. casei*, *L. paracasei*, *L. rhamnosus*, *L. zeae*). The composition of *Bifidobacteria* also changes, with a significant decrease (and even complete absence) of *Bifidobacterium longum* in patients with celiac disease [70,71]. Literature sources indicate an increase in the composition of the intestinal microbiota of *Bacteroides vulgatus* and *Bacteroides fragilis*, which is important because of their gliadin-specific protease activity. It is also noted that virulence and proinflammatory activity are enhanced in some microorganisms, including *Enterobacteriaceae* and *E. coli* isolated from patients [72,73]. In addition to the violation of the taxonomic composition, there is also a change in the metabolic activity of the microflora, manifested in a decrease in the level of short-chain fatty acids in the feces, in particular butyrate [74].

### 2.3. Digestive Proteases

The function of human digestive proteases in the processing of gluten proteins has been extensively researched. The exceptional resistance of glutelin and prolamin to the action of proteolytic enzymes is recognized. This partially decomposed process creates peptides with enterotoxic properties. It is assumed that the presence of a “predisposition” to celiac disease allows these peptides to penetrate the mucous layer, causing a specific inflammatory reaction [28,29,30,41,75,76,77]. Recently, data have become available indicating the direct involvement of the intestinal microbiota in gluten metabolism. This fact needs to be taken into account. In an in vitro experimental study, it was found that bacteria are able to decompose gluten, forming various protein molecules. For example, conditionally pathogenic microorganisms (*Pseudomonas aeruginosa*) were able to produce immunogenic metabolites. Meanwhile, the peptides formed by *Lactobacillus* did not have immunogenic activity, nor did they cause an inflammatory reaction [78]. Thus, active bacteria are present in the microbiota, which can decompose gluten and affect the immunogenicity of gliadin peptides. Research by Caminero A. indicates the presence of 144 bacterial strains of 35 species involved in gluten metabolism. Among them, 94 strains use gluten as the main source of nitrogen, 61 strains exhibit proteolytic activity against gluten proteins and several strains have shown peptidase activity against 33-dimensional peptide (the main immunogenic peptide for patients with celiac disease). Most of these strains belong to the *Firmicutes* and *Actinobacteria* types, mainly to *Lactobacillus, Streptococcus, Staphylococcus, Clostridium* and *Bifidobacterium* [78,79].

In total, it is believed that the human intestine has an extensive variety of bacteria capable of using gluten peptides as nutrients. These bacteria can play an important role in the processing of gluten, preventing the development of celiac disease and presenting new prospects for the treatment of celiac disease. This may include the use of probiotics with preventive and therapeutic effects, as well as possible mitigation of accidental negative effects of gluten exposure.

## 3. Changes in Intestinal Permeability in Celiac Disease

### 3.1. The Intestinal Mucosal–Epithelial Barrier

Intestinal permeability plays a key role in the pathogenesis of gluten enteropathy, which is normally regulated by a multilevel mucosal–epithelial barrier (Figure 1) [80]. The pre-epithelial level of the small intestine is represented by a dense layer of mucus providing a barrier function, which includes antimicrobial peptides, sIgA and glycoproteins [81,82]. The gel-forming mucin MUC2, which is the primary component of mucus in the small intestine, provides protection by inhibiting the adherence of pathogenic bacteria to the intestinal epithelium. While maintaining the balance of microbiota and mucus composition, including normal MUC2 expression, selective impermeability of the pre-epithelial level is maintained [82,83,84].

At the next level—epithelial—there are enterocytes closely connected by a connective complex that maintains the structural integrity of the barrier and regulates the paracellular permeability of the small intestine. This complex includes dense contacts such as tight junction proteins (TJ proteins); transmembrane proteins such as occludin and claudins; peripheral membrane proteins, for example, actin-filament-binding scaffold proteins (ZOs); adhesive molecules (junctional adhesion molecule (JAM), connective adhesion molecule); adhesive contacts (E-cadherin and β-catenin proteins); slit contacts (connexin proteins); and desmosomes [85].

Finally, the deepest layer—the subepithelial layer—is represented by its own plate of the mucous membrane. Thanks to the cells of the immune system (T-lymphocytes, B-lymphocytes, macrophages, dendritic cells), this layer provides immunological protection of the intestinal barrier [86]. The intestinal barrier is a dynamic structure that reacts to the effects of various triggers. Factors such as the nature of nutrition, changes in the microbiota and mucus composition, the level of secretion of inflammatory mediators and hormonal signals affect it [87]. Molecules can penetrate through the intestinal epithelial monolayer in two ways: through the intercellular space (paracellular pathway) or using the transcellular pathway through cells.

### 3.2. The “Leaky” Intestine

It is proved that the violation of the integrity of the epithelial barrier is one of the main etiological factors associated with a number of diseases of the gastrointestinal tract, obesity and diabetes [88]. The violation of the integrity of the mucosal–epithelial barrier in celiac disease is indisputable. However, there is still a debate in the scientific community about whether altered intestinal permeability is the cause or a consequence of immune-mediated reactions. It is obvious that the development of a “leaky” intestine aggravates the nature of the disease [89,90]. The increase in intestinal permeability is due to the interaction of gluten and changes in the microbiota. The composition of the intestinal microbiota influences all aspects of the mucosal–epithelial barrier. In addition, changes in the qualitative and quantitative composition of the intestinal microflora serve as a causal factor in activating the immune system of the intestinal wall with the development of subclinical inflammation, changes in motor function and the development of visceral hypersensitivity, which ultimately leads to disruption of the interaction of the brain–gut–microbiota axis [91,92]. Studies show that an excessive number of Gram-negative bacteria, in particular *Proteobacteria*, is associated with an increase in intestinal permeability and the transfer of microbes, which can become a source of immuno-inflammatory reactions [93,94,95].

How can celiac disease lead to the development of the “flowing” intestinal state? After gluten enters the body, it undergoes decomposition with the aid of intestinal proteases and endopeptidases. Binding of gliadin peptides to the type 3 chemokine receptor (CXCR3) on epithelial cells of the small intestine causes the release of a large amount of zonulin via a signaling pathway dependent on the MyD88 protein (Figure 2). Zonulin, which plays an important role in regulating the permeability of the intestinal wall, activates the actin components of the cytoskeleton of cells associated with zonulin proteins, which leads to an increase in the gaps between cells [41,89,96,97]. Zonulin also enhances intestinal wall permeability by activating the epidermal growth factor receptor (EGFR) via a protease-2 (PAR-2) activated receptor [98]. Disruption of the function of intercellular junctions and the emergence of a “paracellular” pathway for toxic gliadin peptides in the intrinsic plate of the mucous membrane is also associated with an imbalance in the expression of proteins forming a binding structure between enterocytes (decreased expression of E-cadherin, β-catenin and claudins 3 and 4, and increased expression of claudin 2) and cytoskeletal rearrangement. The penetration of gliadin into its own plate of the mucous membrane of the small intestine can also be carried out through the sIgA-CD71 complex using transcytosis [99,100,101].

Gliadin peptides contribute to the activation of the innate immune response in the intestinal epithelium. This response is characterized by an increased expression of IL-15 cytokine in enterocytes. In response, IL-15 enhances the manifestation of an integral membrane protein known as a type 2 activation receptor, which serves as a marker for natural killer cells (NKG2D receptor). This receptor is located inside epithelial lymphocytes. The interaction between MICA proteins (major histocompatibility complex class I polypeptide-related sequence A—associated with the polypeptide of the major histocompatibility complex class I, which is the ligand of the activation receptor of killer cells) and NKG2D receptors on epithelial lymphocytes ultimately leads to enterocyte damage and the initiation of apoptosis [102,103].

Interestingly, the aggravation of gliadin caused an increase in intestinal permeability and this was observed in both patients with celiac disease and in people with gluten sensitivity without celiac disease [104]. In addition, the possibility of using zonulin as a marker was demonstrated [105].

Gliadin peptides penetrate the epithelial barrier and are deamidated by the enzyme tissue transglutaminase (tTG) in the intestinal wall plate. This leads to an increase in binding to HLA-DQ2/DQ8 molecules on the surface of antigen-presenting cells. Such HLA haplotypes play a key role in stimulating the immune response by activating immunogenic gliadin peptides for CD4+ T cells, which triggers the immune cascade. First, stimulation of type 1 T-helper cells leads to the active release of proinflammatory cytokines, such as tumor necrosis factor alpha (TNF-α) and interferon-gamma (IFN-γ), as well as activation of tissue metalloproteinases [106]. Type 2 T-helpers stimulate B-lymphocytes, which causes the formation of specific antibodies to gliadin, tTG and endomysium. As a result of these mechanisms, matrix destruction, mucosal rearrangement, degeneration and death of enterocytes occur, which leads to villi atrophy [93,107,108].

## 4. Effects of Probiotics

### 4.1. Taking Probiotics

*Lactobacillus* and *Bifidobacterium* are typical components of commercial probiotic products. Less popular probiotics are usually based on *Escherichia* or *Saccharomyces*. The main conclusion that can be drawn on the basis of available clinical studies is that the effectiveness of one strain of a microorganism cannot be extrapolated to another strain of this microorganism [109]. The evidence base of the clinical efficacy of probiotics is presented with systematic reviews and meta-analyses of randomized controlled trials. Mosafarybazargany et al. noted that probiotics might alleviate gastrointestinal symptoms, especially in highly symptomatic patients, and improve the immune response in celiac disease and celiac disease autoimmunity patients [110]. Seiler et al. also analyzed seven randomized controlled clinical trials on the effectiveness of probiotics for the treatment of celiac disease, in which probiotics were similarly shown to improve gastrointestinal symptoms in patients with celiac disease [71].

The effects of probiotic microorganisms in celiac disease are studied both in experimental and clinical studies. The beneficial effects of probiotics on the diversity of the intestinal microflora and its main metabolites, including short-chain fatty acids, as well as translocation to other organs of the normoflora are reported [111]. The administration of *Lactobacillus casei* to laboratory animals for 35 days resulted in complete restoration of the villi of the small intestine, reduced weight loss, normalization of basal TNF-α levels and no changes in CD25+ cells and IL-2 levels [112]. Experiments have also shown that the introduction of *Saccharomyces boulardii* and *Bifidobacterium longum* contributes to a decrease in the manifestations of gluten enteropathy, an increase in NFkB and IL-10, as well as a decrease in TNF-α [113,114]. In a systematic review and meta-analysis that included the results of six randomized controlled trials with a total of 5279 participants, it was shown that taking probiotics led to a decrease in gastrointestinal symptoms in patients with celiac disease. The average decrease in symptoms was 228.7% (95% confidence interval (CI) from 213.52 to 243.96; *p* = 0.0002) [115]. In addition, Orlando A. and his colleagues noted that after a 20-day therapy with a probiotic containing *L. rhamnosus*, an increased expression of genes associated with transepithelial slits was observed in patients with celiac disease [115]. In addition, a number of experimental and clinical studies have shown that probiotic bacterial strains can improve the functional integrity of the intestinal epithelial barrier [88]. Moreover, the study of Khorzoghi revealed that the use of multi-strain probiotics at a dosage of three capsules per day for 12 weeks improved the clinical symptoms of patients with celiac disease compared with the placebo group [116].

It should be noted that a large number of probiotic studies based on *Bifidobacterium* have been conducted to solve many health problems, including celiac disease. Apparently, the popularity of *Bifidobacteria* as an object of research is associated with their ability to influence the balance of Th1/Th2-helpers, which is considered a key moment in regulating the activity of the immune system. Thus, *B. bifidum, B. dentium* and *B. longum* are able to stimulate systemic and intestinal immunity [117]. For example, treatment with *B. infantis* has been associated with an improvement in specific symptoms of celiac disease [70]. The possibility of using probiotics based on *Bifidobacterium* as a promising treatment for celiac disease is also emphasized in other publications [62,118,119,120]

### 4.2. The Use of Probiotics for the Breakdown of Gluten in Food

At the moment, various Lactobacilli species capable of decomposing gluten are known—these are *L. ruminus*, *L. john donne*, *L. amylovorus*, *L. salivarius*, *L. alimentaris*, *L. brevis*, *L. sanfranciscenis* and *L. hilgardi*. Studies have shown that if these Lactobacilli species are added to the starter culture for the production of wheat bread, the endopeptidases of these microorganisms are able to decompose gluten peptides. This, in turn, leads to a decrease in the concentration of gluten to levels below 10 ppm (the threshold of gluten-free food) and a decrease in the immunotoxicity of its peptides. In patients suffering from celiac disease and consuming such wheat bread produced with Lactobacilli, there was no worsening of symptoms, increased intestinal permeability or changes in serological markers [121].

In this situation, it is possible to consider the potential modes of action of probiotics from different points of view. Firstly, probiotic microorganisms may have the ability to digest gluten proteins to small non-immunogenic polypeptides, which reduces or eliminates the trigger factor. Secondly, the action of probiotics is aimed at maintaining the permeability of the intestinal barrier, preventing the access of immunogenic polypeptides to the mucous membrane. Finally, probiotics contribute to the restoration of resistance of the intestinal microbiota and the regulation of the innate and adaptive immune system. However, it should be noted that it is not only probiotics that have a positive regulatory effect on the composition of the microbiota, but also a balanced diet rich in vitamins and trace elements [122].

## 5. Conclusions

In sum, scientific studies confirm the changes in the composition of the microbiota in celiac disease, which supports the hypothesis of changes in intestinal bacterial communities. These changes affect the pathogenesis of the disease. It is noted that the intestinal microbiota may have a protective effect on the development of celiac disease. The intestinal microbiota contributes to maintaining the intestinal barrier’s integrity, preventing “leaky” intestine formation. The prescription of probiotics for the treatment and prevention of celiac disease shows encouraging results. The effectiveness of the use of probiotics containing *Lactobacillus* and *Bifidobacterium* bacteria in experimental and clinical studies as a preventive and therapeutic agent has been documented. Preliminary results have proven that the addition of probiotics to a gluten-free diet reduces intestinal hyperpermeability and improves the immune response of the intestine, restoring the normal architecture of the villi. However, additional studies are required to determine the optimal dosage, the choice of strain, the duration of therapy and the start time. These findings require confirmation through randomized clinical trials.

## Figures and Tables

**Figure 1 microorganisms-11-02848-f001:**
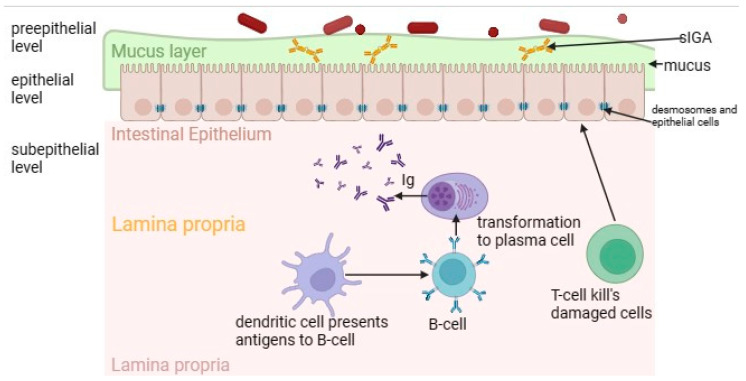
The intestinal mucosal–epithelial barrier in normal circumstances. It consists of pre-epithelial, epithelial and subepithelial levels. The pre-epithelial level is represented by a mucus layer. The epithelial level is represented by enterocytes, which are also closely connected by desmosomes. The subepithelial level includes lamina propria with immune cells.

**Figure 2 microorganisms-11-02848-f002:**
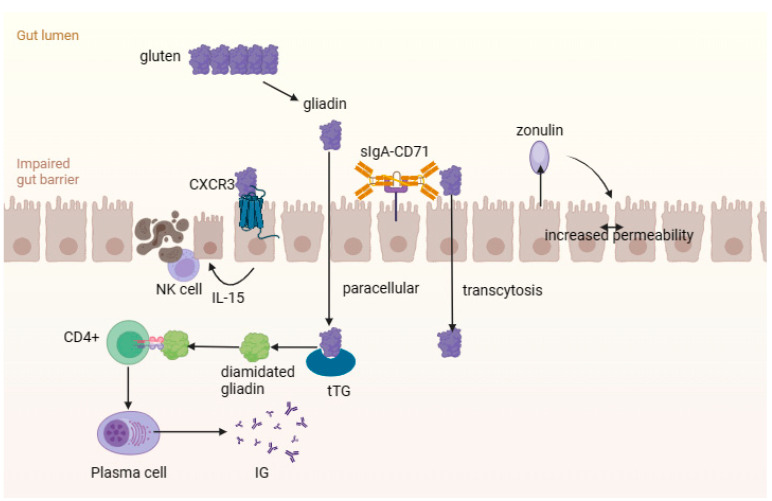
The intestinal mucosal–epithelial barrier in celiac disease. Gluten undergoes decomposition to gliadin peptides, which bind with the type 3 chemokine receptor (CXCR3) and cause the release of zonulin. This increases the gaps between cells and penetration of gliadin into lamina propria. Gliadin peptides contribute to the activation of the innate immune response.

## Data Availability

Not applicable.

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
