# Peer review of "The Contribution of the Intestinal Microbiota to the Celiac Disease Pathogenesis along with the Effectiveness of Probiotic Therapy"

_microorganisms, 2023, doi:10.3390/microorganisms11122848_

Round 1

Reviewer 1 Report

Comments and Suggestions for Authors

This is a concise review of celiac disease. Please make the following modifications.

1.       Please write the bacterial name in italics.

2.       Please explain the abbreviations (HLA, TJ, JAM, ZOs, CO).

3.       Figure 1, Figure 2: Please provide an explanation for the figure. It is unclear from the title alone what the diagram shows. It is also unclear what each illustration represents.

4.       Figure 1: It is unclear how the immune system functions as an intestinal barrier. Furthermore, the relationship between blood vessel and intestinal barrier is unclear.

5.       Figure 2: This diagram is difficult for beginners to understand. It has not been shown what gliadin is derived from, or how zonulin is involved, making it difficult to understand how the intestinal barrier is disrupted. It is also unclear how the sIgA-CD71 complex is involved in gliadin transcytosis. I also think it would be better to clarify that sIgA is an antibody against gliadin.

6.       In the conclusion, it is stated that future challenges will be to clarify the optimal dosage of probiotics, type of strain, duration of administration, and timing of initiation of probiotics for celiac disease. I think it would be better to summarize what kind of research results for these challenges are currently available in Chapter 4.

Author Response

Thank you very much for taking the time to review this manuscript. Please find the detailed responses below and the corresponding revisions in the re-submitted files.

  1. Please write the bacterial name in italics.

Response: We have corrected it.

  1. Please explain the abbreviations (HLA, TJ, JAM, ZOs, CO).

Response: We have explained the abbreviations

  1. Figure 1, Figure 2: Please provide an explanation for the figure. It is unclear from the title alone what the diagram shows. It is also unclear what each illustration represents.

We have provided the explanation.

  1. Figure 1: It is unclear how the immune system functions as an intestinal barrier. Furthermore, the relationship between blood vessel and intestinal barrier is unclear.

Response: We have corrected the figure

  1. Figure 2: This diagram is difficult for beginners to understand. It has not been shown what gliadin is derived from, or how zonulin is involved, making it difficult to understand how the intestinal barrier is disrupted. It is also unclear how the sIgA-CD71 complex is involved in gliadin transcytosis. I also think it would be better to clarify that sIgA is an antibody against gliadin.

Response: We have corrected the diagram

  1. In the conclusion, it is stated that future challenges will be to clarify the optimal dosage of probiotics, type of strain, duration of administration, and timing of initiation of probiotics for celiac disease. I think it would be better to summarize what kind of research results for these challenges are currently available in Chapter 4.

Response: The conclusion has been expanded and also, in chapter 4, we referred to a reference that indicated the dosage and duration of taking probiotics, which led to an improvement in clinical symptoms.

Reviewer 2 Report

Comments and Suggestions for Authors

This paper discusses the contribution of intestinal microbiota to Celiac Disease pathogenesis along with the effectiveness of probiotics therapy. Overall, the paper is well intentioned, but the logic of the article content is rather confusing. We recommend significant revisions to this manuscript.

Some of these errors are listed below:

General information already provided in other literature about intestinal microbiota can be omitted from the part of Introduction. Please refer this reference (Food Chemistry, 417(2023):135861. Critical Reviews in Food Science and Nutrition, Doi: 10.1080/10408398.2023.2179969.).

Please add summarizing words at the beginning of each paragraph that convey the main message of the paragraph. Rather than just listing the contents and results of the experimental study.

The conclusion section can be expanded and added to.

“3. Changes in intestinal permeability in celiac disease”. The role of inflammation should be discussed. Please refer this reference (Critical Reviews in Food Science and Nutrition, Doi: 10.1080/10408398.2022.2076064).

Some other grammatical errors.

Line 17 “are thought to” was suggested with “is thought to”

Line 66 “resent” ?

Line 137 “were infections requiring” was suggested with “was infections requiring”

Line 273 “to endomysium” remove “to”.

The reference should be updated in recent years.

Comments on the Quality of English Language

This paper discusses the contribution of intestinal microbiota to Celiac Disease pathogenesis along with the effectiveness of probiotics therapy. Overall, the paper is well intentioned, but the logic of the article content is rather confusing. We recommend significant revisions to this manuscript.

Some of these errors are listed below:

General information already provided in other literature about intestinal microbiota can be omitted from the part of Introduction. Please refer this reference (Food Chemistry, 417(2023):135861. Critical Reviews in Food Science and Nutrition, Doi: 10.1080/10408398.2023.2179969.).

Please add summarizing words at the beginning of each paragraph that convey the main message of the paragraph. Rather than just listing the contents and results of the experimental study.

The conclusion section can be expanded and added to.

“3. Changes in intestinal permeability in celiac disease”. The role of inflammation should be discussed. Please refer this reference (Critical Reviews in Food Science and Nutrition, Doi: 10.1080/10408398.2022.2076064).

Some other grammatical errors.

Line 17 “are thought to” was suggested with “is thought to”

Line 66 “resent” ?

Line 137 “were infections requiring” was suggested with “was infections requiring”

Line 273 “to endomysium” remove “to”.

The reference should be updated in recent years.

Author Response

Thank you very much for taking the time to review this manuscript. Please find the detailed responses below and the corresponding revisions in the re-submitted files.

General information already provided in other literature about intestinal microbiota can be omitted from the part of Introduction. Please refer this reference (Food Chemistry, 417(2023):135861. Critical Reviews in Food Science and Nutrition, Doi: 10.1080/10408398.2023.2179969.).

Response: We have added the reference in this section.

Please add summarizing words at the beginning of each paragraph that convey the main message of the paragraph. Rather than just listing the contents and results of the experimental study.

Response: We have added subheadings.

The conclusion section can be expanded and added to.

Response: The conclusion has been expanded.

“3. Changes in intestinal permeability in celiac disease”. The role of inflammation should be discussed. Please refer this reference (Critical Reviews in Food Science and Nutrition, Doi: 10.1080/10408398.2022.2076064).

Response: We discussed the aspects of inflammation in the section “3. Changes in intestinal permeability in celiac disease” and referred this reference.

Some other grammatical errors.

Line 17 “are thought to” was suggested with “is thought to”

Response: We have corrected this mistake.

Line 66 “resent” ?

Response: We have corrected this mistake.

Line 137 “were infections requiring” was suggested with “was infections requiring”

Response: We have corrected this mistake.

Line 273 “to endomysium” remove “to”.

Response: We have corrected this mistake.

The reference should be updated in recent years.

Response: We have updated the reference.

Reviewer 3 Report

Comments and Suggestions for Authors

The intestinal microbiota contributes to maintaining the integrity of the intestinal barrier, preventing the formation of a "leaky" intestine. On the contrary, a change in the composition of the microbiota can act as a significant link in the pathogenesis of gluten intolerance and exacerbate the course of the disease. The possibility of modulating the composition of the microbiota by prescribing probiotic preparations is being considered. The effectiveness of the use of probiotics containing Lactobacillus and Bifidobacterium bacteria in experimental and clinical studies as a preventive and therapeutic agent has been documented.

The manuscript is well organized and clearly stated. However, there still have some issues need to be addressed.

1.      Introduction part. Line44-45. Many authors discuss that the method of delivery plays a key role in this process. Please refer this reference (Journal of Agricultural and Food Chemistry. 70(21):6300-6316.).

2.      Celiac disease refers to the inflammation and disturbed of gut microbiota. Please cite the reference about gut microbiota, inflammation, antioxidative effects (Food Chemistry, 402(2023): 134231.).

3.      Please supplement the mechanism about the inflammation in Celiac disease.

4.      The dietary factors should be supplement. Please refer this reference (Comprehensive Reviews in Food Science and Safety, 2023, Doi: 10.1111/1541-4337.13217.).

5.      The expression should be improved.

6.      The reference should be in recent years.

Comments on the Quality of English Language

The intestinal microbiota contributes to maintaining the integrity of the intestinal barrier, preventing the formation of a "leaky" intestine. On the contrary, a change in the composition of the microbiota can act as a significant link in the pathogenesis of gluten intolerance and exacerbate the course of the disease. The possibility of modulating the composition of the microbiota by prescribing probiotic preparations is being considered. The effectiveness of the use of probiotics containing Lactobacillus and Bifidobacterium bacteria in experimental and clinical studies as a preventive and therapeutic agent has been documented.

The manuscript is well organized and clearly stated. However, there still have some issues need to be addressed.

1.      Introduction part. Line44-45. Many authors discuss that the method of delivery plays a key role in this process. Please refer this reference (Journal of Agricultural and Food Chemistry. 70(21):6300-6316.).

2.      Celiac disease refers to the inflammation and disturbed of gut microbiota. Please cite the reference about gut microbiota, inflammation, antioxidative effects (Food Chemistry, 402(2023): 134231.).

3.      Please supplement the mechanism about the inflammation in Celiac disease.

4.      The dietary factors should be supplement. Please refer this reference (Comprehensive Reviews in Food Science and Safety, 2023, Doi: 10.1111/1541-4337.13217.).

5.      The expression should be improved.

6.      The reference should be in recent years.

Author Response

Thank you very much for taking the time to review this manuscript. Please find the detailed responses below and the corresponding revisions in the re-submitted files.

  1. Introduction part. Line44-45. Many authors discuss that the method of delivery plays a key role in this process. Please refer this reference (Journal of Agricultural and Food Chemistry. 70(21):6300-6316.).

Response: We have added the reference in this section.

  1. Celiac disease refers to the inflammation and disturbed of gut microbiota. Please cite the reference about gut microbiota, inflammation, antioxidative effects (Food Chemistry, 402(2023): 134231.).

Response: We have added the reference in this section.

  1. Please supplement the mechanism about the inflammation in Celiac disease.

Response: We have provided it

  1. The dietary factors should be supplement. Please refer this reference (Comprehensive Reviews in Food Science and Safety, 2023, Doi: 10.1111/1541-4337.13217.).

Response: We have added the information about the diet and cited mentioned reference.

  1. The expression should be improved.

Response: The conclusion has been improved and expanded

  1. The reference should be in recent years.

Response: We have updated the reference.

Round 2

Reviewer 1 Report

Comments and Suggestions for Authors

The revised manuscript has been properly modified and become much better. I would appreciate it if you could correct some unedited bacterial names to be italicized.

Good job.

Author Response

Thank you for your comment. We have changed bacterial names.

Reviewer 2 Report

Comments and Suggestions for Authors

The author has responded to the reviewer's comment point by point. It can be accepted in current revision.

Comments on the Quality of English Language

The author has responded to the reviewer's comment point by point. It can be accepted in current revision.

Author Response

We are properly grateful.

Reviewer 3 Report

Comments and Suggestions for Authors

 Please check the reference with the correct issue. It can be accepted in the current revision.

Comments on the Quality of English Language

 Please check the reference with the correct issue. It can be accepted in the current revision.

Author Response

We are properly grateful for your comments.